# A novel mutation in the *SLCO2A1* gene, encoding a prostaglandin transporter, induces chronic enteropathy

Keisuke Jimbo[1], Toshiaki Okuno[2]*, Ryuichi Ohgaki[3], Kou Nishikubo[3], Yuri Kitamura[1], Yumiko Sakurai[1], Lili Quan[3], Hiromichi Shoji[1], Yoshikatsu Kanai[3,4], Toshiaki Shimizu[1], Takehiko Yokomizo[2]

1 Department of Pediatrics, Juntendo University Graduate School of Medicine, Tokyo, Japan, 2 Department of Biochemistry, Juntendo University Graduate School of Medicine, Tokyo, Japan, 3 Department of Biosystem Pharmacology, Graduate School of Medicine, Osaka University, Osaka, Japan, 4 Integrated Frontier Research for Medical Science Division, Institute for Open and Transdisciplinary Research Initiative (OTRI), Osaka University, Osaka, Japan

☯ These authors contributed equally to this work.
* tokuno@juntendo.ac.jp

**Data Availability Statement:** All relevant data are within the manuscript.

**Funding:** This work was supported by MEXT/JSPS KAKENHI grants (16K08596, 19K07357 (to TO), and 15H05904, 15H05897, 18H02627, and

## Abstract

Chronic enteropathy associated with *SLCO2A1* gene (CEAS) is caused by loss-of-function mutations in *SLCO2A1*, which encodes a prostaglandin (PG) transporter. In this study, we report a sibling case of CEAS with a novel pathogenic variant of the *SLCO2A1* gene. Compound heterozygous variants in *SLCO2A1* were identified in an 8-year-old boy and 12-year-old girl, and multiple chronic nonspecific ulcers were observed in the patients using capsule endoscopy. The splice site mutation (c.940 + 1G>A) of the paternal allele was previously reported to be pathogenic, whereas the missense variant (c.1688T>C) of the maternal allele was novel and had not yet been reported. The affected residue (p.Leu563Pro) is located in the 11th transmembrane domain (helix 11) of SLCO2A1. Because SLCO2A1 mediates the uptake and clearance of PGs, the urinary PG metabolites were measured by liquid chromatography coupled to tandem mass spectrometry. The urinary tetranor-prostaglandin E metabolite levels in the patients were significantly higher than those in unaffected individuals. We established cell lines with doxycycline-inducible expression of wild type SLCO2A1 (WT-SLCO2A1) and the L563P mutant. Immunofluorescence staining showed that WT-SLCO2A1 and the L563P mutant were dominantly expressed on the plasma membranes of these cells. Cells expressing WT-SLCO2A1 exhibited time- and dose-dependent uptake of $PGE_2$, while the mutant did not show any uptake activity. Residue L563 is very close to the putative substrate-binding site in SLCO2A1, R561 in helix 11. However, in a molecular model of SLCO2A1, the side chain of L563 projected outside of helix 11, indicating that L563 is likely not directly involved in substrate binding. Instead, the substitution of Pro may twist the helix and impair the transporter function. In summary, we identified a novel pathogenic variant of *SLCO2A1* that caused loss-of-function and induced CEAS.

19KK0199 (to TY)); the Mishima Kaiun Memorial Foundation (to TO) and the Ono Medical Research Foundation (to TY).The funders had no role in study design, data collection and analysis, decision to publish, or preparation of the manuscript.

**Competing interests:** The authors have declared that no competing interests exist.

## Introduction

Chronic enteropathy associated with *SLCO2A1* gene (CEAS), which was originally called chronic nonspecific multiple ulcers of the small intestine (CNSU), is caused by loss-of-function mutations in the solute carrier organic anion transporter family, member 2A1 (*SLCO2A1*) gene [1]. Human *SLCO2A1* is located on chromosome 3q22.1-q22.2 and consists of 14 exons that encode a cell-surface prostaglandin (PG) transporter comprising 643 amino acids and 12 transmembrane (TM) domains [2]. Prostanoids, which are synthesized from arachidonic acid (AA) via cyclooxygenase (COX)-mediated pathways, are released from cells and bind to their cognate receptors to enact their biological activities [3]. Prostanoid signaling is terminated by cellular uptake of prostanoids, followed by their intracellular oxidation by 15-hydroxyprostaglandin dehydrogenase (15-PGDH) [4, 5], and thus, this uptake is a critical step in prostanoid inactivation [6, 7].

It is generally accepted that chemical or genetic inhibition of SLCO2A1 elevates tissue concentrations of prostaglandin $E_2$ ($PGE_2$), resulting in augmentation of $PGE_2$ signaling through PGE receptors (EPs) [8]. Clinical studies with individuals bearing recessive mutations in *SLCO2A1* have demonstrated that *SLCO2A1* is a causative gene for primary hypertrophic osteoarthropathy (PHO) and CEAS, in which $PGE_2$ catabolism is defective [1, 9]. CEAS is an autosomal recessive inherited disease [1], and the multiple ulcers that form in the small intestine cause the chronic loss of blood and protein in these patients [10, 11]. Previously, CEAS was diagnosed based on clinical symptoms, including intestinal edema, abdominal pain, and gastrointestinal bleeding, and the identification of distinctive intestinal ulcers using capsule endoscopy or balloon-assisted endoscopy was required to distinguish CEAS from intestinal Crohn's disease [12, 13]. However, given that CEAS has been shown to be driven by SLCO2A1 loss-of-function [1, 11], the disease is diagnosed by clinical features and a genetic test in the *SLCO2A1* gene. In this study, we report a sibling case of CEAS with a novel mutation in the *SLCO2A1* gene that resulted in loss-of-function of $PGE_2$ uptake by SLCO2A1.

## Results

### Identification of compound heterozygous variants in *SLCO2A1* gene

A 12-year-old girl (patient III-1) and 8-year-old boy (patient III-3) were born to healthy non-consanguineous parents (Fig 1A). Initially, the boy exhibited hypoproteinemia and severe anemia, and was referred to a former institution. No remarkable past medical and consanguineous marriage history were found in his parents. Because the fecal occult blood test was positive, esophagogastroduodenoscopy and colonoscopy were performed, but we could not confirm obvious mucosal abnormalities. However, we observed multiple circular and tape-like shallow ulcers in the entire small intestine and predominantly in the lower ileum using intestinal capsule endoscopy, strongly indicating CEAS (Fig 1B). We identified compound heterozygous variants in the *SLCO2A1* gene using direct sequencing. The splice site mutation (c.940 + 1G>A) of the paternal allele was previously reported [14], whereas the missense variant (c.1688T>C) of the maternal allele was novel and had not been reported before (Fig 1C). The affected residue (p.Leu563Pro) is located in helix 11 of SLCO2A1 (Fig 1D). Screening of SLCO2A1 orthologs using the NCBI HomoloGene database (http://www.ncbi.nlm.nih.gov/sites/entrez?cmd=Retrieve&db=homologene&dopt=MultipleAlignment&list_uids=38077) revealed that the Leu563 residue is completely conserved among humans, chimpanzees, monkeys, dogs, cows, mice, rats, chickens, and frogs (Fig 1D), which suggests that the missense mutation is possibly pathogenic. Through genetic analysis of the parents and siblings, we

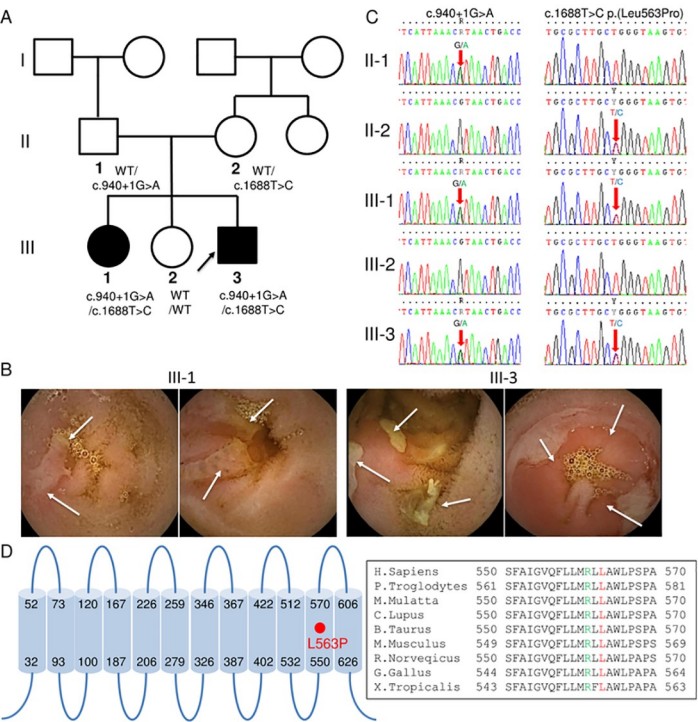

**Fig 1. Pedigree of the chronic enteropathy associated with SLCO2A1 (CEAS) family with a novel mutation of *SLCO2A1* gene.** (**A**) The segregation of the *SLCO2A1* mutations c.940 + 1G>A (splice site) and c.1688 T>C (L563P) is indicated. Squares represent male family members, circles represent female family members, and black symbols represent clinically affected family members. (**B**) Capsule endoscopic findings. Multiple shallow tape-like ulcers (patient III-1) and multiple circular and tape-like shallow ulcers (patient III-3) were observed in the small intestinal mucous, respectively. (**C**) Trio-based sequence chromatogram of the *SLCO2A1* mutations. (**D**) Schematic diagram representing the position of L563P in the 11th transmembrane helix. L563 in the helix is highly conserved among different species. R561 is the putative substrate-binding site.

identified the same heterozygous variants in the *SLCO2A1* gene of patient III-1, who had past medical history of recurrent abdominal pain, but it had not been scrutinized at medical institutes. After the identification of the genetic mutation, she was found to have hypoproteinemia and anemia similar to patient III-3 by laboratory test. She also exhibited multiple shallow and tape-like ulcers in entire small intestine and predominantly in the lower ileum by capsule endoscopy (Fig 1B). In addition, because *SLCO2A1* defects also cause another disease primary hypertrophic osteoarthropathy (PHO)/pachydermoperiostosis (PDP), we consulted two CEAS patients to a dermatologist in our hospital and got the answer that there were no characteristic findings of PHO/PDP.

## The urinary tetranor-prostaglandin metabolites levels

As SLCO2A1 mediates the uptake and clearance of PGs, the urinary PG metabolites of the patients were measured using liquid chromatography coupled to tandem mass spectrometry (LC-MS/MS). Tetranor-prostaglandin E metabolite (t-PGEM) is a major metabolite of $PGE_2$ and is used as a urinary marker of $PGE_2$ biosynthesis (Fig 2A) [15]. The levels of urinary t-PGEM and tetranor-prostaglandin F metabolite (t-PGFM) in the CEAS patients were significantly higher than those in unaffected individuals, whereas tetranor-prostaglandin D metabolite (t-PGDM) levels were relatively comparable (Fig 2B).

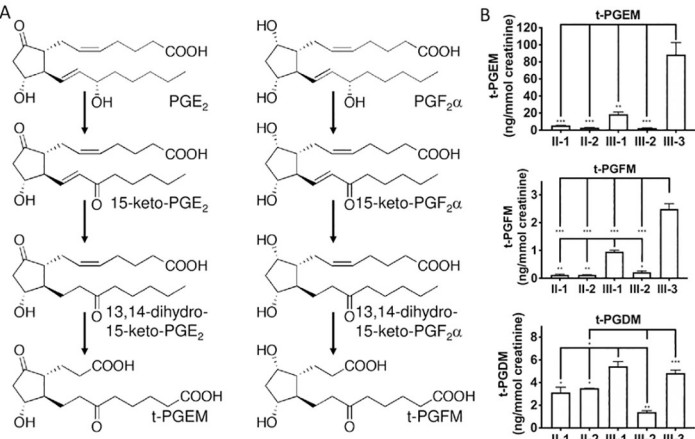

**Fig 2. Urinary levels of tetranor-prostaglandin metabolites, t-PGEM, t-PGFM, and t-PGDM.** (**A**) Metabolic pathways of prostaglandin $E_2$ and prostaglandin $F_2\alpha$. (**B**) Urinary t-PGEM, t-PGFM, and t-PGDM were measured using UPLC-MS/MS. Each bar represents the mean ± SEM (n = 3). ***, $p < 0.0001$; **, $p < 0.001$; *, $p < 0.01$ (One-way ANOVA and Tukey's multiple comparisons test).

## Functional analysis of WT-SLCO2A1 and L563P mutants

For functional analysis of the newly identified *SLCO2A1* mutation, we established Flp-In T-Rex 293 cell lines with doxycycline-inducible expression of the intact or Leu563Pro (L563P) mutated SLCO2A1. Western blot analysis revealed that the expressions of wild type (WT)-SLCO2A1 and the L563P mutant were comparable (Fig 3A), and immunofluorescence

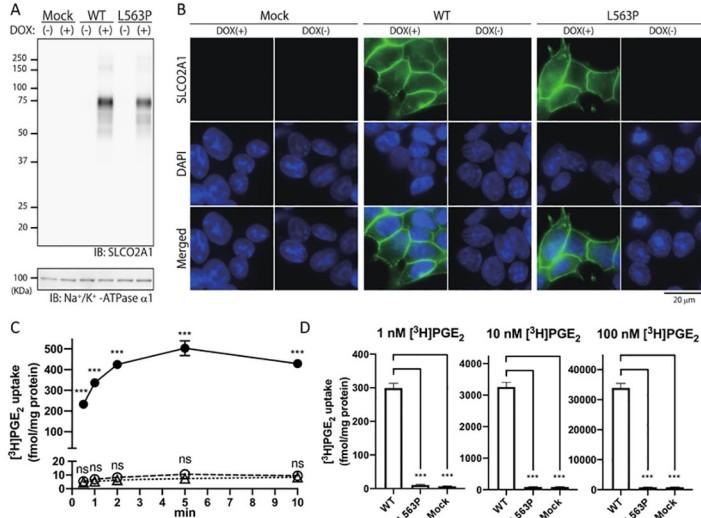

**Fig 3. Expression, subcellular localization, and PGE$_2$ uptake activities of SLCO2A1 and the L563P mutant.** (**A**) Membrane fractions of Flp-In T-REx 293 cells expressing WT-SLCO2A1 and the L563P mutant were subjected to western blotting analysis. (**B**) Cells were treated with doxycycline (DOX) for 24 hr and fixed. Immunofluorescence staining was performed with anti-SLCO2A1 antibody and Alexa Fluor 488-conjugated goat anti-rabbit IgG. (**C**) Cells were treated with DOX, and then incubated for the indicated periods with 1 nM [$^3$H]PGE$_2$. After extensive washing with HBSS, the cells were lysed with 0.1 N NaOH and the radioactivity was measured. Data represent the mean ± SEM (n = 4). ***, $p < 0.0001$, 'ns', not significant. (One-way ANOVA and Tukey's multiple comparisons test). (**D**) Cells were treated with DOX and incubated for 1 min with 1, 10, and 100 nM [$^3$H]PGE$_2$. The radioactivity was measured as described above. The uptake values were normalized by the protein concentrations in the cell lysates. Experiments were repeated three times, with quadruplicates for each sample (n = 4). Each bar represents the mean ± SEM. ***, $p < 0.0001$ (One-way ANOVA and Tukey's multiple comparisons test).

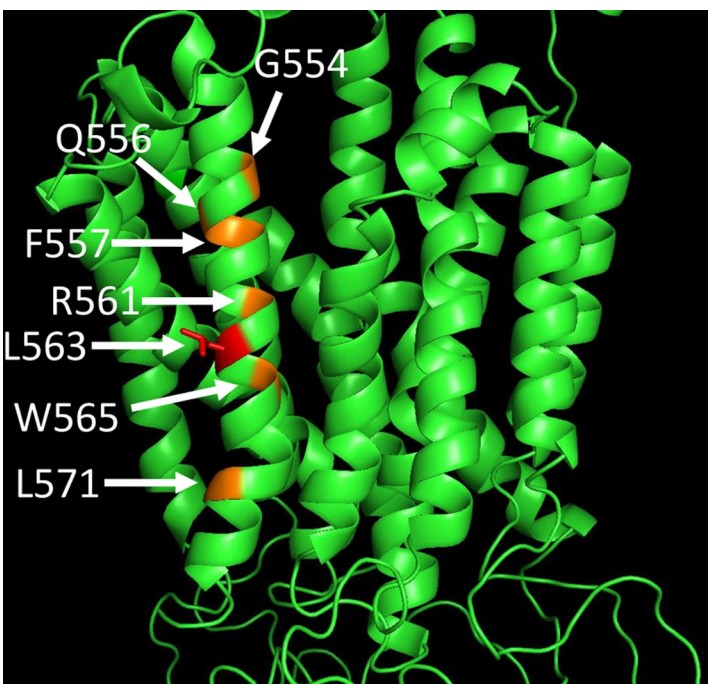

**Fig 4. Molecular model of SLCO2A1.** A molecular model of SLCO2A1 was constructed using the crystal structure of a glycerol-3-phosphate transporter. The Leu563 in the 11th transmembrane helix is shown in red. The mutations reported in the text are displayed in orange.

staining showed that WT-SLCO2A1 and the L563P mutant were both dominantly expressed on the plasma membrane (Fig 3B). Cells expressing WT-SLCO2A1 displayed time-dependent uptake of $^3$H-labeled PGE$_2$ ([11] PGE$_2$), but the mutant did not show any uptake activity at all (Fig 3C). In an assay measuring the transport rate of PGE$_2$, cells expressing WT-SLCO2A1 showed significant [$^3$H]PGE$_2$ uptake in a dose-dependent manner, while the L563P mutation resulted in complete disruption of SLCO2A1 activity at 1, 10, and even 100 nM PGE$_2$ (Fig 3D).

## The structure model of SLCO2A1

To illustrate the effects of the missense mutation in helix 11 of SLCO2A1, a molecular model of SLCO2A1 was constructed using the crystal structure of a glycerol-3-phosphate transporter (GlpT, PDB ID:1PW4) [16]. The molecular model of SLCO2A1 was constructed with the SWISS-MODEL server and Swiss-PdbViewer from the template PDB, and was analyzed using PyMol. The substrate may occupy a pocket between the transmembrane helixes similar to that of GlpT. The L563 residue is located close to the putative substrate-binding site at R561, and the L563 side chain projects outward from helix 11 (Fig 4). This suggests that while the L563 side chain is not directly involved in substrate binding, its substitution with proline may twist the helix and impair the transporter function.

## Discussion

In the present study, we report a novel pathogenic variant of SLCO2A1 identified in a sibling case of CEAS. Among the known *SLCO2A1* mutations, a splice-site mutation at intron 7 (c.940 + 1G>A; rs765349238) is the most frequently observed among CEAS patients, with 54% of mutant allele frequency. This mutation results in the deletion of the entire exon 7 from SLCO2A1, leading to a frameshift at amino acid position 288 and the creation of a premature

stop codon after six amino acid residues (p.R288Gfs*7) [1]. Based on the data from the Human Genetic Variation Database (HGVD) [17], the allele frequency of c.940 + 1G>A in the Japanese population is 0.091% (2/2188). The missense variant (c.1688T>C) identified in this work has not previously been identified, and the affected residue (p.Leu563Pro) is located in the 11th transmembrane domain of SLCO2A1. Several mutations, including G554R, R561C, L571P [14], Q556H [18], F557S [19], and W565G [20], have been reported in the 11th transmembrane domain. SLCO2A1 belongs to the superfamily of 12-transmenbrane organic anion transporting polypeptides (OATPs) [6–8], and R561 and K614 in the 11th and 12th transmembrane domains, respectively, are thought to be involved in ligand binding [8, 21]. The novel variant of L563 is close to R561, and may influence the ligand binding activity of the transporter (Fig 4).

PGE$_2$ is degraded through two main steps. First, prostaglandin transporters (PGTs), including SLCO2A1, SLCO3A1, and SLCO4A1, mediate PGE$_2$ cellular uptake, which allows it to subsequently be oxidized to 15-keto-PGE$_2$ by 15-PGDH [22] and then degraded to t-PGEM [23]. The levels of PGE$_2$ and t-PGEM in the serum and urine of CNSU patients are significantly higher than in control groups [14, 24]. Because SLCO2A1 is highly expressed in the lungs [2, 25] and is localized to vascular endothelial cells, SLCO2A1 expression in lung vascular cells may contribute to PGE$_2$ uptake [26]. Patient III-3 (8-year-old boy, Fig 1A), with hypoproteinemia and severe anemia, showed the highest t-PGEM levels, while patient III-1 (12-year-old girl, Fig 1A), with milder symptoms, showed the second highest t-PGEM level in this family. These results suggest that the t-PGEM level may reflect the severity of the disease in patients. The urinary levels of t-PGFM and t-PGDM were much lower than those of t-PGEM, but the ratio between these metabolites in patients and healthy controls was similar (Fig 2). This can be explained by the fact that SLCO2A1 can transport PGF$_2\alpha$ and PGD$_2$, in addition to PGE$_2$ [2], and the degradation pathways of PGF$_2\alpha$ and PGD$_2$ are similar to PGE$_2$.

*Slco2a1*-knockout mice were previously generated and analyzed [26, 27], and the impact of the *SLCO2A1* mutation on the plasma t-PGEM levels was different in these organisms was different compared to humans. The urinary t-PGEM levels of *Slco2a1*-deficient mice and human patients with CEAS were higher than in the control mice and healthy controls, respectively. In contrast, while the plasma t-PGEM levels of *Slco2a1*-deficient mice were lower than in control mice [27], the plasma t-PGEM levels in CEAS patients were higher than in healthy controls [9]. This might be due to differences in the PGE$_2$ metabolic activities between mice and humans. Further, although recent studies have shown that *SLCO2A1* function is lost in patients with CEAS, *Slco2a1*-deficient mice do not spontaneously develop enteritis [28]. This might be because *Slco2a1/SLCO2A1* has different functions in mice and humans. Alternatively, patients with CEAS might have a "second-hit factor," other than a lack of *SLCO2A1*, such as other genetic factors or certain pathogens in the intestinal lumen. Considering the clinical impact of these differences, this mouse model may not be the best tool for analyzing CEAS, which mainly affects the small intestine.

PGE$_2$ concentrations in patients with CEAS should theoretically be higher than in those without this mutation because of reduced PGE$_2$ metabolism. However, the t-PGEM concentrations in patients with CEAS have been shown to be higher than those in patients with Crohn's disease [29]. Measuring t-PGEM is useful for distinguish CEAS from Crohn's disease, although the roles of PGE$_2$ in enteritis remain uncertain.

An *in vitro* overexpression experiment showed that the expression of WT-SLCO2A1 and the L563P mutant was comparable (Fig 3A), and that both the WT-SLCO2A1 and the L563P mutant were dominantly expressed on the plasma membrane (Fig 3B). These results suggest that the L563P mutation did not affect the stability and localization of the protein, but instead impaired the transport function of SLCO2A1. In previous reports, cells transiently expressing

the transporter were used for PGE$_2$ uptake assays; however, these studies did not address the expression and the cellular localization of the transporter [1, 20]. The transporter expression on the plasma membrane requires multiple steps, including protein folding and membrane sorting. Our strategy will be useful for analyzing the molecular functions of mutant SLCO2A1 transporters with missense mutations.

In summary, we identified a novel mutation in SLCO2A1 (L563P) that affects the function of this transporter in PGE$_2$ transport and causes CEAS.

## Material and methods

### Ethics statement

Written informed consent for the studies was obtained from each individual. Written statement that formal consent was obtained from the parent. The study was approved by the Juntendo University Research Ethics Committee.

### Genetic analysis

Genomic DNA was extracted from the peripheral leukocytes of the patients and their parents using a standard procedure with QIAamp DNA Blood Mini Kit (QIA). All 14 exons and exon–intron splice junction regions of the *SLCO2A1* gene were amplified by PCR (Ampli Taq Gold ABI, USA and Thermal Cycler C1000 Touch Bio-Rad). PCR products were purified and cycling sequencing was performed using Big Dye terminator Mix V1.1 (ABI) for direct DNA sequence analysis (Genetic Analyzer 3500, ABI). Sequences were analyzed using Seq Scape software V.2.5 (ABI). The reference sequence data is based on GenBank NG_031964.2. Primers for mutation analysis of the *SLCO2A1* gene are shown in S1 Table.

### Urinary prostaglandin metabolites analysis

The measurements of t-PGEM, t-PGFM, and t-PGDM were performed as previously described [30]. An aliquot of urine (20 µL) was diluted with 100 µL of methanol and incubated for 1 hr at -20˚C. After the addition of 1 mL of water, stable isotope-labeled internal standards, t-PGEMd6, and t-PGDMd6, the solution was loaded onto Oasis HLB cartridges (Waters Corporation, Milford, MA, USA). The column was sequentially washed with water containing 0.1% formic acid (Solvent A), 5% methanol containing 0.1% formic acid, and petroleum ether containing 0.1% formic acid, and the samples were eluted with 200 µL of methanol containing 0.1% formic acid. For reverse-phase ultra-high performance liquid chromatography coupled to tandem mass spectrometry (RP-UPLC-MS/MS), we used the ACQUITY UPLC H-Class system (Waters Corporation) and a TSQ Quantum Ultra triple quadrupole mass spectrometer, equipped with an electrospray ionization ion source (Thermo Fisher Scientific, Walford, MA, USA). An aliquot of each sample (20 µL) was diluted with 30 µL of Solvent A and analyzed by RP-UPLC-MS/MS using an ACQUITY UPLC HSS C18 column (2.1 × 100 mm, 1.8 µm particles) (Waters Corporation). The column temperature was set at 41˚C, and the samples were separated at a flow rate of 250 µL/min with a linear gradient from 5% to 30% of Solvent B (acetonitrile) in three minutes, and then 100% in 15 minutes. The samples were then kept at 100% for two minutes before re-equilibration for two minutes. The settings for the mass spectrometer were as follows: ESI voltage of 2500 V, tube lens of 110 V, collision energy of 13 V, capillary temperature 225˚C, and vaporizer temperature of 250˚C. Mass spectrometric analysis was performed in the negative ion mode using selective reaction monitoring, and the transitions that were monitored were: 327.2 > 309.2 m/z (-H$_2$O) for t-PGEM and t-PGDM, 333.2 > 315.2 m/z (-H$_2$O) for t-PGEM-d6 and t-PGDM-d6, respectively, and 329.1 > 311.1 m/z (-H$_2$O) for t-

PGFM. Automated peak detection, calibrations, and calculations were carried out using the Xcalibur 2.2 software package (Thermo Fisher Scientific).

The absorbance of urinary creatinine was measured using a Creatinine (urinary) Colorimetric Assay Kit (Cayman Chemical, Ann Arbor, Michigan). The values for t-PGEM, t-PGDM, and t-PGFM are presented as ng/mg creatinine.

## Construction of expression vectors

Expression vectors for pcDNA3.1(+)-WT-SLCO2A1 (NM_005630.2) and the L563P mutant were constructed by VectorBuilder (Chicago, IL). The DNA fragments containing a Kozak sequence and full-length coding sequence were excised by HindIII/ApaI-digestion, and subcloned into pcDNA5/FRT/TO (Invitrogen-Thermo Fisher Scientific, Waltham, MS). Obtained expression vectors were used to establish doxycycline-inducible stable cell lines.

## Construction of stable cell lines

Doxycycline-inducible stable cell lines of WT-SLCO2A1 and the L563P mutant using Flp-In T-REx 293 cells were constructed and maintained as previously described [31]. Empty vector pcDNA5/FRT/TO was used to generate control cell lines (Mock). Obtained clones were tested for the expression of SLCO2A1 protein in crude membrane fractions by western blot analysis. Briefly, cells were seeded in 10 cm dish ($4\times10^6$ cells/dish) containing DMEM high glucose w/o phenol red (Wako Pure Chemicals, Osaka, Japan), supplemented with 10% (v/v) heat-inactivated FBS (Gibco Life Technologies, Grand Island, NY), 100 units/mL penicillin (Nacalai Tesque, Kyoto, Japan), 100 μg/mL streptomycin (Nacalai Tesque, Kyoto, Japan), 150 μg/mL hygromycin B (Wako Pure Chemicals, Osaka, Japan), and 5 μg/mL Blasticidin (InvivoGen, San Diego, CA). 24 hr after the passage, doxycycline (Clontech, Mountain View, CA) was added to the medium to a final concentration of 1 μg/mL, and the cells were further incubated for 24 hr. The cells were collected on ice in ice-cold PBS and lysed by sonication in lysis buffer containing 20 mM Tris-HCl (pH 7.5), 150 mM NaCl, 1 mM EDTA, and protease inhibitor cocktail (Roche Diagnostics GmBH, Mannheim, Germany). Whole cell lysate was centrifuged at $1,000 \times g$ for 5 min, and the supernatant was further centrifuged at $541,000 \times g$ for 1 hr to obtain crude membrane fraction. After solubilization with the lysis buffer containing 10% (v/v) glycerol and 1% (v/v) NP-40, samples were mixed with Laemmli buffer. SDS-PAGE followed by western blot analysis was conducted as previously described [32]. Primary antibodies used were: anti-SLCO2A1 (HPA013742, Sigma-Aldrich, St. Louis, MO), and anti-Na$^+$/K$^+$ ATPase α1 (sc-21712, Santa Cruz Biotechnology, Santa Cruz, CA). The immunoreactive band intensity of SLCO2A1 was quantified by densitometry using Image J software (National Institutes of Health, Bethesda, MD). Clones expressing similar levels of WT and the L563P mutant were selected for PGE$_2$ uptake assays and immunofluorescence staining.

## Immunofluorescence staining

Cells were seeded in 3.5 cm dishes ($1\times10^6$ cells/dish) containing collagen-I-coated coverslips, and were treated with doxycycline as described above. Immunofluorescence staining and image acquisition was performed as described previously [33]. Antibodies used were: anti-SLCO2A1 (HPA013742, Sigma-Aldrich, St. Louis, MO) as the primary antibody, and Alexa Fluor 488-conjugated goat anti-rabbit IgG (A11034, Molecular probes-Thermo Fisher Scientific, Waltham, MS) as the secondary antibody.

## PGE$_2$ uptake assay

Cells were seeded on poly-D-lysine coated 24-well plates (1×10$^5$ cells/well), and treated with doxycycline as described above. After preincubation for 5 min in Hanks' balanced salt solution (HBSS: 125 mM NaCl, 4.8 mM KCl, 1.2 mM MgSO$_4$, 1.2 mM KH$_2$PO$_4$, 1.3 mM CaCl$_2$, 5.6 mM D-glucose, and 25 mM HEPES [pH 7.4]), cells were incubated for the indicated periods with 1 nM [$^3$H]PGE$_2$ ([5,6,8,11,12,14,15-$^3$H(N)]-PGE$_2$, 1.85 MBq/mmol, PerkinElmer, Waltham, MA) to perform a time course analysis. To test the concentration dependence of uptake, cells were incubated for 1 min with 1, 10, and 100 nM [$^3$H]PGE$_2$. After extensive washing with HBSS, the cells were lysed with 0.1 N NaOH and radioactivity was measured by liquid scintillation counting as previously described [34]. The uptake values were normalized by protein concentration of cell lysates determined using a BCA Protein Assay Kit (Thermo Fisher Scientific, Waltham, MS). Experiments were repeated three times, with quadruplicates for each sample (n = 4).

## Statistics

One-way ANOVA and Tukey's multiple comparisons test was used to determine *p* values. *P* values $\leq 0.05$ were considered statistically significant. The statistical analyses were performed with GraphPad Prism 8 (GraphPad Software, Inc., San Diego, CA).

## Supporting information

**S1 Table. Primers for mutation analysis of the SLCO2A1 gene.**
(DOCX)

## Acknowledgments

We thank the patients and their family for participating in this study. We also thank Dr. Yasushi Okazaki for valuable comments and Dr. Kyoko Yasuda for technical assistance. We would like to thank Enago (www.enago.jp) for the English language review.

## Author Contributions

**Conceptualization:** Toshiaki Okuno, Takehiko Yokomizo.

**Funding acquisition:** Toshiaki Okuno.

**Investigation:** Keisuke Jimbo, Toshiaki Okuno, Ryuichi Ohgaki, Kou Nishikubo, Yuri Kitamura, Yumiko Sakurai, Lili Quan, Hiromichi Shoji.

**Supervision:** Yoshikatsu Kanai, Toshiaki Shimizu, Takehiko Yokomizo.

**Writing – original draft:** Keisuke Jimbo, Toshiaki Okuno, Ryuichi Ohgaki.

**Writing – review & editing:** Toshiaki Okuno, Takehiko Yokomizo.

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
