## [Decision Letter · Decision Letter 0]

22 Sep 2020

PONE-D-20-25461

A novel mutation in the SLCO2A1 gene, encoding a prostaglandin transporter, induces chronic enteropathy

PLOS ONE

Dear Dr. Okuno,

Thank you for submitting your manuscript to PLOS ONE. After careful consideration, we feel that it has merit but does not fully meet PLOS ONE’s publication criteria as it currently stands. Therefore, we invite you to submit a revised version of the manuscript that addresses the points raised during the review process.

We look forward to receiving your revised manuscript.

Kind regards,

Sylvie Mazoyer, Ph.D.

Academic Editor

PLOS ONE

Journal Requirements:

Reviewers' comments:

Reviewer's Responses to Questions

**Comments to the Author**

1. Is the manuscript technically sound, and do the data support the conclusions?

Reviewer #1: Yes

Reviewer #2: Yes

2. Has the statistical analysis been performed appropriately and rigorously? 

Reviewer #1: Yes

Reviewer #2: Yes

3. Have the authors made all data underlying the findings in their manuscript fully available?

Reviewer #1: Yes

Reviewer #2: Yes

4. Is the manuscript presented in an intelligible fashion and written in standard English?

Reviewer #1: Yes

Reviewer #2: Yes

5. Review Comments to the Author

Reviewer #1: Jimbo et al. reported a sibling case with chronic enteropathy associated with SLCO2A1 gene (CEAS) and found a new missense variant (c.1688T>C) in the SLCO2A1 gene. They also confirmed that this mutation causes loss of transporter function by the PGE2 uptake experiment.

Major point

1. Clinical features about the patients are unclear. For example, there is no description of patient III-1 and the location of the small intestinal lesions of patient III-3 is unclear. They should describe in detail the gastrointestinal and extraintestinal features of the patients. If possible, they should discuss the difference between siblings.

Minor point

1. P6L74. The authors wrote, "the disease can now be genetically diagnosed...". but CEAS is diagnosed by clinical features and a genetic test and genetic test just help in diagnosing. Please revise this sentence.

2. P6L81. The word "consanguineous" should be wrong. Please make sure.

3. P9L120. The word "CNSU" should be changed to CEAS.

4. P10L150. There is no explanation about the representation of the black circle and so on in the Fig 3c.

Reviewer #2: This is a new case report for the sibling of CEAS with compound heterozygous variants (c.940+1G>A and c.1688T>C). Jibo et al found a novel SLCO2A1 mutation (c.1688T>C) and confirmed that the SLCO2A1 variant (L563P) had no PGE2 transport activity when it was expressed in Flp-In T-Rex 293 cells. The structure model of SLCO2A1 suggests the substitution of Leu with Pro impairs the transport function. This paper is well-described, and molecular mechanism supports their clinical observation. Information on clinically associated mutations of SLCO2A1 is worthwhile to be shared for efficient genetical diagnosis. The following minor concerns are raised.

1. Figure 2A shows the metabolic pathway of both PGE2 and PGF2a; however, they are basically the same. They should combine them and show only essential parts of the pathway.

2. SLCO2A1 defects also cause PHO/PDP. Have the authors observed any clinical features of PHO/PDP in these patients? It would be appreciated to describe or mention clinical manifestations of PHO/PDP.

3. In Figure 4, the authors suggest that L563 side chain is not directly involved in substrate biding because the side chain is outward (line 168). Does this mean that the side of the leucine outward from the helix does not affect substrate binding? This description is a little bit hard to understand. It would be nicer If the authors clarified this molecular basis more clearly.

6. PLOS authors have the option to publish the peer review history of their article (what does this mean?). If published, this will include your full peer review and any attached files.

Reviewer #1: **Yes: **Junji Umeno

Reviewer #2: No

---

## [Author Response · Author response to Decision Letter 0]

14 Oct 2020

Point-by-point Responses

Reviewer #1: Jimbo et al. reported a sibling case with chronic enteropathy associated with SLCO2A1 gene (CEAS) and found a new missense variant (c.1688T>C) in the SLCO2A1 gene. They also confirmed that this mutation causes loss of transporter function by the PGE2 uptake experiment.

Major point

1. Clinical features about the patients are unclear. For example, there is no description of patient III-1 and the location of the small intestinal lesions of patient III-3 is unclear. They should describe in detail the gastrointestinal and extraintestinal features of the patients. If possible, they should discuss the difference between siblings.

Response: Thank you very much for your critical comments. We described clinical features about the patients as follows:

P6L86: However, we observed multiple circular and tape-like shallow ulcers in the entire small intestine and predominantly in the lower ileum using intestinal capsule endoscopy, strongly indicating CEAS (Fig. 1B).

P7L98: Through genetic analysis of the parents and siblings, we identified the same heterozygous variants in the SLCO2A1 gene of patient III-1, who had past medical history of recurrent abdominal pain, but it had not been scrutinized at medical institutes. After the identification of the genetic mutation, she was found to have hypoproteinemia and anemia similar to patient III-3 by laboratory test. She also exhibited multiple shallow and tape-like ulcers in entire small intestine and predominantly in the lower ileum by capsule endoscopy (Fig. 1B). 

Minor point

1. P6L74. The authors wrote, "the disease can now be genetically diagnosed...". but CEAS is diagnosed by clinical features and a genetic test and genetic test just help in diagnosing. Please revise this sentence.

Response: Thank you for your critical comment. We revised our manuscript as follow.

P6L74: the disease is diagnosed by clinical features and a genetic test in the SLCO2A1 gene.

2. P6L81. The word "consanguineous" should be wrong. Please make sure.

Response: Thank you for your useful comment. We modified our manuscript as follow.

P6L81: to healthy non-consanguineous parents (Fig. 1A).

3. P9L120. The word "CNSU" should be changed to CEAS.

Response: Thank you for your useful comment. We modified our manuscript as follow.

P9L127: in the CEAS patients were significantly higher

4. P10L150. There is no explanation about the representation of the black circle and so on in the Fig 3c.

Response: Thank you for your useful comment. We added the representations of the black circle, open circle, and open triangle in the Fig 3c.

We thank Reviewer #1 for the constructive and insightful comments, which helped us to substantially improve our manuscript.

Reviewer #2: This is a new case report for the sibling of CEAS with compound heterozygous variants (c.940+1G>A and c.1688T>C). Jibo et al found a novel SLCO2A1 mutation (c.1688T>C) and confirmed that the SLCO2A1 variant (L563P) had no PGE2 transport activity when it was expressed in Flp-In T-Rex 293 cells. The structure model of SLCO2A1 suggests the substitution of Leu with Pro impairs the transport function. This paper is well-described, and molecular mechanism supports their clinical observation. Information on clinically associated mutations of SLCO2A1 is worthwhile to be shared for efficient genetical diagnosis. The following minor concerns are raised.

1. Figure 2A shows the metabolic pathway of both PGE2 and PGF2a; however, they are basically the same. They should combine them and show only essential parts of the pathway.

Response: Thank you for useful comments. We have combined the metabolic pathway of PGE2 and PGF2a in Figure 2A.

2. SLCO2A1 defects also cause PHO/PDP. Have the authors observed any clinical features of PHO/PDP in these patients? It would be appreciated to describe or mention clinical manifestations of PHO/PDP.

Response: Thank you for critical comments. We consulted two CEAS patients to a dermatologist in our hospital and got the answer that there were no characteristic findings of PHO/PDP. Therefore, we modified our manuscript as follows.

P7L104: (Fig. 1B). In addition, because SLCO2A1 defects also cause another disease primary hypertrophic osteoarthropathy (PHO)/pachydermoperiostosis (PDP), we consulted two CEAS patients to a dermatologist in our hospital and got the answer that there were no characteristic findings of PHO/PDP.

3. In Figure 4, the authors suggest that L563 side chain is not directly involved in substrate biding because the side chain is outward (line 168). Does this mean that the side of the leucine outward from the helix does not affect substrate binding? This description is a little bit hard to understand. It would be nicer If the authors clarified this molecular basis more clearly.

Response: Thank you for useful comments. We consider that the side chain of the leucine outward from the helix is not involved in substrate binding. We modified the manuscript as follows:

P12L172: The substrate may occupy a pocket between the transmembrane helixes similar to that of GlpT. 

We thank Reviewer #2 for the constructive and insightful comments, which have helped us to substantially improve our manuscript.

---

## [Decision Letter · Decision Letter 1]

22 Oct 2020

A novel mutation in the SLCO2A1 gene, encoding a prostaglandin transporter, induces chronic enteropathy

PONE-D-20-25461R1

Dear Dr. Okuno,

We’re pleased to inform you that your manuscript has been judged scientifically suitable for publication and will be formally accepted for publication once it meets all outstanding technical requirements.

Kind regards,

Sylvie Mazoyer, Ph.D.

Academic Editor

PLOS ONE

Additional Editor Comments (optional):

Reviewers' comments:

Reviewer's Responses to Questions

**Comments to the Author**

1. If the authors have adequately addressed your comments raised in a previous round of review and you feel that this manuscript is now acceptable for publication, you may indicate that here to bypass the “Comments to the Author” section, enter your conflict of interest statement in the “Confidential to Editor” section, and submit your "Accept" recommendation.

Reviewer #2: All comments have been addressed

2. Is the manuscript technically sound, and do the data support the conclusions?

Reviewer #2: Yes

3. Has the statistical analysis been performed appropriately and rigorously? 

Reviewer #2: Yes

4. Have the authors made all data underlying the findings in their manuscript fully available?

Reviewer #2: Yes

5. Is the manuscript presented in an intelligible fashion and written in standard English?

Reviewer #2: Yes

6. Review Comments to the Author

Reviewer #2: Additional information about clinical features in these CEAS patients is highly appreciated and useful.

7. PLOS authors have the option to publish the peer review history of their article (what does this mean?). If published, this will include your full peer review and any attached files.

Reviewer #2: **Yes: **Takeo Nakanishi

---

## [Editor Report · Acceptance letter]

29 Oct 2020

PONE-D-20-25461R1 

A novel mutation in the *SLCO2A1* gene, encoding a prostaglandin transporter, induces chronic enteropathy 

Dear Dr. Okuno:

I'm pleased to inform you that your manuscript has been deemed suitable for publication in PLOS ONE. Congratulations! Your manuscript is now with our production department. 

Kind regards, 

on behalf of

Dr Sylvie Mazoyer 

Academic Editor

PLOS ONE